# On anomalies and gauging of U(1) non-invertible symmetries in 4d QED

Avner Karasik

Department of Applied Mathematics and Theoretical Physics,
University of Cambridge, CB3 0WA, UK

avnerkar@gmail.com

## Abstract

In this work we propose a way to promote the anomalous axial $U(1)$ transformations to exact non-invertible $U(1)$ symmetries. We discuss the procedure of coupling the non-invertible symmetry to a (dynamical or background) gauge field. We show that as part of the gauging procedure, certain constraints are imposed to make the gauging consistent. The constraints emerge naturally from the form of the non-invertible $U(1)$ conserved current. In the case of dynamical gauging, this results in new type of gauge theories we call non-invertible gauge theories: These are gauge theories with additional constraints that cancel the would-be gauge anomalies. By coupling to background gauge fields, we can discuss 't-Hooft anomalies of non-invertible symmetries. We show in an example that the matching conditions hold but they are realized in an unconventional way. Turning on non-trivial background for the non-invertible gauge field changes the vacuum even when the symmetry is not broken and the background is very weak. The anomalies are then matched by the appearance of solitons in the new vacuum.

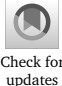

# 1  Introduction

One of the fascinating recent developments in theoretical physics is the notion of non-invertible symmetries. See for example [1–15] for a partial list of references. Symmetries play an extremely important role in physics. Given a physical system, the first thing we should do is to identify and classify all the symmetries of the theory. This is obviously true for ordinary symmetries, but also for special symmetries, such as the so-called non-invertible symmetries. Once we found the symmetries, the next question we should ask our selves is what can we do with it. For ordinary invertible symmetries the answer is well known. It includes conservation laws, selection rules, anomaly matching conditions, Goldstone bosons and many more. We can also gauge ordinary symmetries assuming they don't suffer from anomalies. Is the same true for non-invertible symmetries? In this note we would like to study the non-invertible version of two important concepts: Gauging and anomaly matching conditions [16]. Let us start by briefly reminding how this works for ordinary symmetries. Given a theory $\mathcal{T}$ with a global symmetry group $G$, we can add gauge fields $A$ for $G$. At this point we don't integrate over the values of $A$ but choose to study the theory in a specific background $A(x)$. It happens to be that even though $G$ is an exact symmetry of the theory, once we add $A$, the partition function is not necessarily invariant under the action of $G$ but satisfies,

$$\mathcal{Z}[A] \rightarrow e^{iS[\omega,A]}\mathcal{Z}[A]. \tag{1}$$

$S[\omega,A]$ is a local action of the gauge parameters $\omega$ and the gauge fields $A$ that cannot be removed by adding local counter terms. $e^{iS[\omega,A]}$ is an overall phase that can be taken out of the path-integral. Equation (1) is RG invariant in the sense that the same relation must be satisfied by the partition function at any scale. The action $S[\omega,A]$ is the anomaly action and the conditions (1) are the 't-Hooft anomaly matching conditions. If $G$ (or a subgroup of it) is anomaly free, i.e. $S[\omega,A] = 0$, we can consistently integrate over all the values of $A(x)$. Now the gauge fields $A$ become dynamical field in the theory. This dynamical gauging cannot be done if $G$ suffers from anomalies. The reason is that when we gauge a symmetry, we identify all the configurations that differ by the action of the symmetry. This can be done consistently only if all these configurations are really equivalent. In the case of anomalies, configurations that differ by the action of the symmetry are note equivalent- they differ by the phase $e^{iS[\omega,A]}$, and therefore it makes no sense to identify them. To understand how to do it for non-invertible symmetries, we first need to understand what non-invertible symmetries are. It will be useful to first explain how to define symmetries using the terminology of topological operators [17]. Having a p-form symmetry is equivalent to having an operator $U_g(\mathcal{M}_{d-p-1})$ defined on some $d - p - 1$ manifold $\mathcal{M}_{d-p-1}$. This operator is topological- it can be twisted and deformed, and as long as it doesn't cross an insertion of another operator, the theory is invariant. This operator is also unitary. In particular, there always exists an inverse operator $U_{g^{-1}}$ such that if we bring the two of them together we get the identity operator. Since the operator is unitary, we can always write it as $U_g(\mathcal{M}_{d-p-1}) = e^{i \int_{\mathcal{M}_{d-p-1}} \star J_g}$ with a well defined current $J_g$. While this construction is very general, we will focus on 0-form $U(1)$ symmetries for simplicity. In

this case we can write $U_\alpha(\mathcal{M}_{d-1}) = e^{i\alpha \int_{\mathcal{M}_{d-1}} \star J}$ with $d \star J = 0$. These operators are topological, unitary and obviously invertible. They are well defined when acting on any operator or state in the Hilbert space. Now we will move on to non-invertible symmetries. We will describe here a way to construct non-invertible symmetries that works in several cases. Let's assume that we found a topological unitary operator $U$ that has only one (not so minor) downside- Its action is not well defined on every state in the Hilbert space. The conservative approach to this operator is that it is illegal and we should ignore it. A bolder approach is to say that this operator still generates a well defined symmetry on a subspace and we can try to combine $U$ with a projection operator $P$ that projects to the subspace on which $U$ is well defined. The combined operator $\tilde{U} = U \cdot P$ is well defined but isn't unitary nor invertible.

As a concrete realization, let's assume that we are able to write a current $J$ which is conserved $d \star J = 0$, but not gauge invariant. Under gauge transformations parametrized by $g$, the current goes to $J \to J_g$ with $d \star J_g = 0$. So actually we have a family of conserved non-gauge invariant currents $\{J_g\}$. We can define the operator (up to normalization)

$$\tilde{U}_\alpha(\mathcal{M}_{d-1}) = \sum_g \exp\left( i\alpha \int_{\mathcal{M}_{d-1}} \star J_g \right). \tag{2}$$

$\tilde{U}$ is a sum of topological operators, and hence topological itself. It is gauge invariant, since gauge transformations just reshuffle the different terms in the sum. However, it is in general not unitary and non-invertible. We conclude that $\tilde{U}$ generates a non-invertible symmetry. Through out the paper we will use the $\tilde{\ }$ symbol to denote objects related to non-invertible symmetries (operators, currents, gauge fields...). Effectively, the sum over gauge transformations projects us to the subspace on which $U_\alpha = e^{i\alpha \int \star J}$ is gauge invariant.

This construction may seem too simple to work. Indeed, there are cases (see section 5) where the sum over $g$ in (2) is so drastic such that $\tilde{U}$ is identically 0. Still, there are cases in which $\tilde{U}$ is non-trivial and generates a true non-invertible symmetry. The case we will mainly focus on in this work is $U(1)$ gauge theories in four spacetime dimensions. These theories contain an anomalous axial current [18, 19], satisfying $d \star j = \frac{k}{8\pi^2} da \wedge da$ where $a$ is the dynamical $U(1)$ gauge field and $k$ is an integer that depends on the details of the theory. We can define the current $\star J = \star j - \frac{k}{8\pi^2} a \wedge da$ which is conserved but not gauge invariant. Under gauge transformations,

$$a \to a - d\phi \quad \Rightarrow \quad \star J \to \star\tilde{J} = \star J + \frac{k}{8\pi^2} d\phi \wedge da. \tag{3}$$

Therefore we can define the operator

$$\tilde{U}_\alpha(\mathcal{M}_3) = \mathcal{D}\phi \ \exp\left( i\alpha \int_{\mathcal{M}_3} \left[ \star j - \frac{k}{8\pi^2} a \wedge da + \frac{k}{8\pi^2} d\phi \wedge da \right] \right). \tag{4}$$

Summing over gauge transformations is equivalent to adding an auxiliary compact scalar living only on $\mathcal{M}_3$. We claim that this operator generates a non-invertible $U(1)$ symmetry. Recently, the authors of [13, 14] proposed adding a new Chern-Simons gauge field that lives on $\mathcal{M}_3$ and argued that this gives rise to a well defined non-invertible symmetry for every rational $\alpha$. While the two operators may look different, we claim they are completely equivalent. Similar to the scalar $\phi$, also the auxiliary gauge field of [13,14] effectively projects us to the subspace on which $U_\alpha$ is gauge invariant. The two operators are equivalent for every rational $\alpha$. The advantage of the approach presented here is that the definition of (4) can be easily extended to any real value of $\alpha$ and therefore results in a continuous $U(1)$ symmetry, instead of a discrete subgroup. This construction makes it easier to gauge the non-invertible $U(1)$ symmetry and

study its 't-Hooft anomalies as will be shown. The main results can be summarized as follows. We know that the anomalous $U(1)$ generated by $j$ above is not a symmetry of the theory due to the ABJ anomaly. As such, it cannot be coupled to gauge fields- background or dynamical- and doesn't give rise to rigorous 't-Hooft anomaly matching conditions. It's non-invertible version is an exact symmetry and therefore we can try and couple it to a gauge field $\tilde{b}$. As part of the gauging procedure, we add to the Lagrangian the term

$$\tilde{b} \wedge \star \tilde{J} = \tilde{b} \wedge \star J + \frac{k}{8\pi^2} \tilde{b} \wedge d\phi \wedge da \,. \tag{5}$$

Notice that once we introduce the gauge field $\tilde{b}$, $\phi$ becomes a 4d field that lives everywhere in space-time. $\phi$ appears only in (5) and acts as a Lagrange multiplier enforcing the constraint

$$d\tilde{b} \wedge da = 0 \,. \tag{6}$$

This constraint holds no matter if $\tilde{b}$ is a background or a dynamical gauge field. Even if $\tilde{b}$ is background, we still path-integrate over $\phi$. The constraint is a crucial ingredient in gauging the non-invertible symmetry. It eliminates all the "bad" configurations that would have made the gauging inconsistent. Explicitly, we will see that this constraint eliminates all the would-be gauge anomalies giving rise to new type of gauge theories we call non-invertible gauge theories. The constraint also plays an important role in 't-Hooft anomaly matching conditions for the non-invertible symmetries. In particular, when studying the theory in a specific $\tilde{b}$ background, the constraint modifies the dynamics of the theory and drives the theory to a different vacuum from the one with $\tilde{b} = 0$. In the new vacuum, the anomalies that can be observed by the choice of $\tilde{b}$ are matched.

The outline of the paper is as follows. In section 2 we construct explicitly the non-invertible $U(1)$ symmetry and discuss some of its general properties. We will also comment about the similarities between our construction and the construction of [13,14]. In section 3 we gauge the $U(1)$ dynamically and show that we get a constrained gauge theory. In section 4 we study anomalies of the non-invertible symmetry by coupling it to background gauge fields. In section 5 we explain why the procedure of 2 fails to work in other cases. Section 6 is a more general discussion about the relation between non-invertible symmetries and the ideas of [20]. These are two apparently different methods to promote the anomalous axial $U(1)$ to an exact symmetry and use this symmetry to learn new things about the dynamics of the theory. We argue that in some sense the two methods are equivalent and give two completing ways to look at the axial symmetry.

## 2 Non-invertible axial $U(1)$ in QED

We will start by introducing a way to redefine the anomalous axial $U(1)$ in QED to be an exact non-invertible symmetry. Related strategies were introduced in [13,14]. Consider $U(1)$ gauge theory with a gauge field $a_\mu$, and a charged Dirac fermion $\psi$ with charge 1. Under an axial transformation of the fermion, $\psi \to e^{i\alpha\gamma_5/2}\psi$, the action is modified by

$$\delta S = \frac{\alpha}{8\pi^2} \int da \wedge da \,, \quad \frac{1}{8\pi^2} \int da \wedge da \in \mathbb{Z} \,, \tag{7}$$

such that the transformation is trivial for $\alpha = 2\pi\mathbb{Z}$. The axial current $j_\mu = \frac{1}{2}\bar{\psi}\gamma_5\gamma_\mu\psi$ is not conserved but satisfies $d \star j = \frac{1}{8\pi^2} da \wedge da$. We can instead define the current

$$\star J = \star j - \frac{1}{8\pi^2} a \wedge da \,, \tag{8}$$

which is conserved. However, it is not gauge invariant. Instead of looking at the current, we can look at the operator generating the symmetry,

$$U_\alpha = e^{i\alpha \int \star J} = \exp\left( i\alpha \int \star j - \frac{i\alpha}{8\pi^2} \int a \wedge da \right), \tag{9}$$

where the integration is over some 3-manifold $\mathcal{M}_3$. This operator is topological- this is a direct consequence of the fact that the current is conserved. However, it is gauge invariant only when the coefficient of the Chern-Simons action is $\frac{1}{4\pi}\mathbb{Z}$. This is true when $\alpha = 2\pi\mathbb{Z}$ which is the same condition we had before. But for $\alpha = 2\pi\mathbb{Z}$ the operator acts on the fermion as $\psi \to -\psi$ which is gauge equivalent to the identity. Can we find a way to redefine the symmetry to get something gauge invariant and topological for non-trivial values of $\alpha$? The problem with $U_\alpha$ is that it is not gauge invariant. Under gauge transformations,

$$a \to a - d\phi : \; U_\alpha \to U_\alpha^\phi = e^{i\alpha \int \star \tilde{J}}, \quad \star\tilde{J}(\phi) = \star J + \frac{1}{8\pi^2} d\phi \wedge da. \tag{10}$$

If we sum over all the values of $\phi$, we get a gauge invariant operator,

$$\tilde{U}_\alpha = \int \mathcal{D}\phi \, e^{i\alpha \int \star \tilde{J}}. \tag{11}$$

Here $\int \mathcal{D}\phi$ is a suitably normalized path integral over all the configurations of $\phi$ on $\mathcal{M}_3$. This procedure guarantees gauge invariance, as well as conservation. This is true since $d \star J_\phi = 0$ for every $\phi$. An alternative way to interpret this procedure is that we add new degrees of freedom living only on $\mathcal{M}_3$. The idea to add new degrees of freedom in such a way to make the axial $U(1)$ an exact symmetry was used before. In [20] it has been shown that by adding new heavy fields, it is possible to make a $\mathbb{Z}_N$ discrete subgroup of the axial symmetry an exact symmetry of the theory. The effect of the heavy fields on the low energy theory is in the emergence of Chern-Simons terms on $\mathbb{Z}_N$ domain walls. This construction can give new non-trivial constraints on the low energy theory similar to conventional anomaly matching conditions. A little bit later, a similar approach has been taken in [13,14] where it has been shown that the operator $U_\alpha$ can become gauge invariant by adding Chern-Simons gauge fields living on it. It was shown that the modified $U_\alpha$ is non-invertible. In particular, it acts on fermions simply by axial rotations, but annihilates 't-Hooft lines. Also in this approach, only a subgroup of the axial $U(1)$ can be restored. We propose to add instead a compact scalar $\phi$ on $\mathcal{M}_3$. The advantage is that the new current $\tilde{J}$ is conserved locally and therefore the operator $\tilde{U}_\alpha$ is topological for every value of $\alpha$.

Let's try to understand better some of the properties of the operator, which we write here in full glory,

$$\tilde{U}_\alpha(\mathcal{M}_3) = \int \mathcal{D}\phi \; \exp\left[ i\alpha \int_{\mathcal{M}_3} \left( \star j - \frac{1}{8\pi^2} a \wedge da + \frac{1}{8\pi^2} d\phi \wedge da \right) \right]. \tag{12}$$

As we already said, this operator is gauge invariant for every value of $\alpha$. Another way to view it is by defining the covariant derivative of $\phi$ as $D\phi = d\phi - a$. Then the current can be written in a manifestly gauge invariant way,

$$\star\tilde{J} = \star j + \frac{1}{8\pi^2} D\phi \wedge da. \tag{13}$$

$\tilde{U}_\alpha$ is topological since the current is conserved,

$$d\left( \star j - \frac{1}{8\pi^2} a \wedge da + \frac{1}{8\pi^2} d\phi \wedge da \right) = 0. \tag{14}$$

This equation requires some clarification. In particular, $\phi$ is defined as a 3d field living only on $\mathcal{M}_3$, so what does it mean to take its derivative in the direction orthogonal to $\mathcal{M}_3$? In general, it will not be possible to extend $\phi$ to th entire 4d spacetime without singularities. However, this is not needed. When we continuously deform the manifold $\mathcal{M}_3$ to $\mathcal{M}_3'$ we cover a 4d manifold with the topology of $\mathcal{M}_3 \times I$, where $I$ is an interval. All we need is to be able to extend $\phi$ from $\mathcal{M}_3$ to $\mathcal{M}_3 \times I$ and this is always possible. Therefore, (14) is well defined, Stokes theorem can be used safely and $\tilde{U}_\alpha(\mathcal{M}_3)$ is indeed topological.

However, $\tilde{U}_\alpha$ is not unitary. This is due to the path-integration over $\phi$. What is the effect of this $\phi$ and the additional term $d\phi \wedge da$?

This term is a total derivative. It is completely trivial on compact $\mathcal{M}_3$ with no non-trivial holonomies. On $S^2 \times S^1$ for example, we can replace the path integration over $\phi$ with a discrete sum over the holonomies[1]

$$\tilde{U}_\alpha(\mathcal{M}_3) \sim \sum_k exp\left[ i\alpha \int_{\mathcal{M}_3} \left( \star j - \frac{1}{8\pi^2} a \wedge da \right) + \frac{i\alpha k}{2\pi} \int_{S^2} da \right]. \tag{15}$$

For irrational $\alpha$, the sum over $k$ results in the no-flux condition, $\int_{S^2} da = 0$ since

$$\sum_k \exp\left[ \frac{i\alpha k}{2\pi} \int_{S^2} da \right] \sim \delta_{\int_{S^2} da, 0}. \tag{16}$$

If $\alpha$ is rational, we can write it as $\alpha = \frac{p}{N}$ where $p, N$ are coprime integers. The sum over $k$ in this case gives a weaker constraint: $\int da \in 2\pi N\mathbb{Z}$. At the end of the day, we can write our operator as

$$\tilde{U}_\alpha = \exp\left[ i\alpha \int \left( \star j - \frac{1}{8\pi^2} a \wedge da \right) \right] P_\alpha, \tag{17}$$

where $P_\alpha$ is a projection operator on the subspace of allowed fluxes. However, the presentation of the operator using the scalar $\phi$ turns out to be useful when coupling this symmetry to gauge fields as we will see in the next sections.

The projection operator is a sign for non-invertibility: The operator $\tilde{U}$ acts as 0 on certain configurations.

How does $\tilde{U}_\alpha$ act on operators of the theory? On the fermions it acts as axial rotation by $\alpha$. No surprises here. On line operators the story is more subtle. Consider as an example $\mathcal{M}_3$ to be defined by $t = -\epsilon$ and a Wilson line on $t = y = z = 0$. As we take $\mathcal{M}_3$ to $t = +\epsilon$, the commutation relations of the two objects will give us the action of $\mathcal{D}$ on the Wilson line. Since there is no time component involved in any of them, the commutator is trivially zero and the operator doesn't act on Wilson lines. Now, replace the Wilson line with a 't-Hooft line. This is done by replacing the gauge field $a$ with the dual gauge field $a^D$. While $a_x$ has non-trivial commutation relations with $\partial_t a_x$, the dual gauge field $a_x^D$ has non-trivial commutation relations with $f_{yz}$. We find that the action of this operator on a 't-Hooft line is

$$e^{i \int a^D} \to \int \mathcal{D}\phi \, e^{i \int (a^D - \alpha a/(2\pi) + \alpha d\phi/(2\pi))}. \tag{18}$$

If $\alpha \in 2\pi\mathbb{Z}$, the operator just takes a magnetic line to a dyonic line with integer electric charge of $q_E = \frac{1}{2\pi}\alpha$ as expected from the Witten effect [21, 22]. The (appropriately normalized) integration over $\phi$ in this case is trivial,

$$\int \mathcal{D}\phi \, e^{iq_E \int d\phi} = 1. \tag{19}$$

---

[1]There is a tricky factor of 2 involved in this computation. The origin of this factor of 2 is explained in appendix A.

For general $\alpha$, the electric charge of the dyonic line is fractional which is not allowed. Any way, the integration over $\phi$ for $\alpha \neq 2\pi\mathbb{Z}$ gives zero so we are saved from getting the fractional dyon. We see that the operator $\tilde{U}_\alpha$ annihilates 't-Hooft line operators. This is another sign for its non-invertibility. This behaviour is equivalent to that of the topological operator introduced in [13, 14]. In fact, the two definitions of the topological operators give the same result when acting on all the physical states and operators of the theory. We will show explicitly equivalence for the case where $\alpha = \frac{2\pi}{N}$. From (17), we can write our operator as

$$\tilde{U}_{2\pi/N}(\mathcal{M}_3) = \exp\left[\frac{2\pi i}{N}\int_{\mathcal{M}_3}\left(\star j - \frac{1}{8\pi^2}a\wedge da\right)\right]P_{2\pi/N}, \tag{20}$$

where $P_{2\pi/N}$ is zero when $\int da \neq 0 \mod 2\pi N$ on any closed two dimensional submanifold of $\mathcal{M}_3$. The operator of [13, 14] can be written as

$$D_{2\pi/N}(\mathcal{M}_3) = \int \mathcal{D}A \exp\left[i\int_{\mathcal{M}_3}\left(\frac{2\pi}{N}\star j + \frac{N}{4\pi}A\wedge dA + \frac{1}{2\pi}A\wedge da\right)\right]. \tag{21}$$

Here $A$ is the gauge field living only on $\mathcal{M}_3$. The generalization to $D_{2\pi p/N}$ with $gcd(p, N) = 1$ is somewhat subtle. For our purpose, what we need to know is that $D_{2\pi p/N}$ has a $\mathbb{Z}_N$ 1-form symmetry with anomaly $p$, and that $da/N$ plays the role of a background $\mathbb{Z}_N$ 2-form gauge field coupled to the $\mathbb{Z}_N$ 1-form symmetry. A consequence of the anomaly is that,

$$D_{2\pi p/N}(\mathcal{M}_3) = \exp\left(\frac{ip}{2\pi N}\int_{\mathcal{M}_3}d\omega\wedge da\right)D_{2\pi p/N}. \tag{22}$$

This equation must hold for any scalar $\omega$ satisfying $\oint d\omega \in 2\pi\mathbb{Z}$. If there is $\omega$ such that the phase is non-trivial, $D_{2\pi p/N} = 0$. It is non-zero only when $\int da \in 2\pi N\mathbb{Z}$ on any closed two dimensional submanifold of $\mathcal{M}_3$. In this case, the defect field $A$ can be safely integrated out resulting in

$$D_{2\pi/N}(\mathcal{M}_3) \rightarrow \exp\left[\frac{2\pi i}{N}\int_{\mathcal{M}_3}\left(\star j - \frac{1}{8\pi^2}a\wedge da\right)\right]. \tag{23}$$

This is identical to (20). We conclude that for every rational $\alpha$, the two definitions are identical. The operator (21) doesn't have a generalization to irrational $\alpha$. On the other hand, (12) is defined for every real $\alpha$.

Next we will discuss some of the applications of the symmetry. In particular, how to gauge the non-invertible symmetry in section 3 and how to derive anomaly matching conditions in section 4.

## 3   Non-invertible gauge theories

We will start this section by explaining how to couple $\tilde{U}(1)$ to a gauge field. In the case of ordinary continuous symmetries, we have a current $J$ and the minimal coupling of the symmetry to a gauge field $b$ simply involves adding $b_\mu J^\mu$ to the Lagrangian. We can re-derive this result using topological operators. Coupling to a gauge field $b(x)$ is equivalent to inserting into the path integral many copies of the topological operator $U_\alpha(\mathcal{M}_3) = e^{i\alpha\int_{\mathcal{M}_3}\star J}$ on various choices of 3-manifolds $\mathcal{M}_3$. Every set of insertions $\{U_\alpha(\mathcal{M}_3)\}$ corresponds to a choice of vector field $b(x)$ as can be seen in figure 1. The result is that the insertion of topological operators is equivalent to adding the minimal coupling term $b_\mu J^\mu$ to the Lagrangian. This procedure can

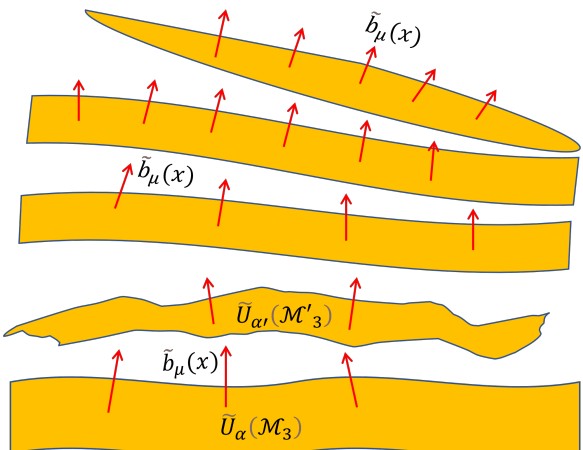

**Figure 1:** The set of insertions of topological operators $\tilde{U}$ can be described by a vector field $\tilde{b}_\mu$ orthogonal to the manifolds on which $\tilde{U}$ are defined. Effectively, inserting into the path integral $\left\langle \tilde{U}_\alpha(\mathcal{M}_3)\tilde{U}_{\alpha'}(\mathcal{M}'_3)...\right\rangle$ is equivalent to inserting $\left\langle \int D\phi \, e^{i\int \tilde{b}_\mu \tilde{J}^\mu}\right\rangle$ where $\phi$ now lives in the entire 4d space-time.

be easily generalized to the non-invertible symmetry we're talking about. From the point of view of topological operators, the procedure is exactly the same. This procedure tells us how to couple this non-invertible symmetry to a gauge field $\tilde{b}$, and the way to do it is to add to the Lagrangain the minimal coupling term $\tilde{b}_\mu \tilde{J}^\mu$ and integrate over $\phi$ as a 4d field. See figure 1 for more details.

This differs in two ways from the naive and inconsistent attempt of coupling the anomalous axial symmetry to a gauge field. First, $\tilde{b}_\mu J^\mu$ contains a cubic interaction term between the gauge fields, $\delta\mathcal{L}_{int} = -\frac{1}{8\pi^2}\tilde{b}\wedge a\wedge da$. Second, $\tilde{b}_\mu J^\mu$ contains the term $d\phi \wedge \tilde{b}\wedge da$. This is the only place where $\phi$ appears. We can integrate $\phi$ out which results in the local constraint

$$d\tilde{b}\wedge da = 0. \tag{24}$$

The constraint and the interaction together make sure that the theory is well-defined without gauge anomalies. We will go over several examples.

## 3.1 Non-invertible $U(1)_a \times \tilde{U}(1)_b$ gauge theory

Consider as a first example a theory with 4 Weyl fermions $\psi_i$ and the two symmetries $U(1)_{a,b}$ with the charges[2]

$$q_1 = (1,1), \; q_2 = (-1,1), \; q_3 = (0,-1), \; q_4 = (0,-1). \tag{25}$$

Notice that the anomalies $U(1)^3_{a,b}$, $U(1)_{a,b}\times gravity$, $U(1)_a\times U(1)^2_b$ vanish. The only anomaly that doesn't vanish is $U(1)^2_a \times U(1)_b$. Due to this anomaly we cannot gauge the two $U(1)$ symmetries simultaneously. Instead, we can gauge $U(1)_a$ first, and then promote the anomalous $U(1)_b$ to the non-invertible symmetry $\tilde{U}(1)_b$ and gauge it. Denote the two gauge fields for $U(1)_a \times \tilde{U}(1)_b$ by $a, \tilde{b}$ respectively. After gauging $U(1)_a$, the non-invertible $\tilde{U}(1)_b$ current becomes

$$\star\tilde{J}_b = \star j_b - \frac{1}{8\pi^2}a\wedge da + \frac{1}{8\pi^2}d\phi\wedge da, \tag{26}$$

---

[2]We ignore the other symmetries of the theory and treat them as accidental as they won't play any role in our construction.

where $j_b$ is the classical anomalous current. As explained above, when we gauge $\tilde{U}(1)_b$ we add the term $\tilde{b}_\mu J^\mu \supset \frac{1}{8\pi^2}\tilde{b} \wedge d\phi \wedge da$. $\phi$ acts as a Lagrange multiplier. Integrating over $\phi$ we get the constraint $d\tilde{b} \wedge da = 0$. We see that we can gauge two $U(1)$ symmetries with a mixed anomaly. The price that we pay is the constraint $da \wedge d\tilde{b} = 0$ which saves us from the gauge anomaly. This theory is well defined, but we don't integrate over the entire $U(1)_a \times \tilde{U}(1)_b$ configuration space but only over a subspace. The Lagrangian for this $U(1)_a \times \tilde{U}(1)_b$ gauge theory is

$$\mathcal{L} = \frac{1}{4g_a^2}(da)^2 + \frac{1}{4g_b^2}(d\tilde{b})^2 + i\sum_{j=1}^4 \psi_j^\dagger \sigma^\mu D_\mu \psi_j - \frac{1}{8\pi^2}\tilde{b} \wedge a \wedge da \,. \tag{27}$$

This theory is different from a naive $U(1) \times U(1)$ gauge theory in two ways. First, the term $\tilde{b} \wedge a \wedge da$ appears to make the theory gauge invariant under $\tilde{U}(1)_b$ gauge transformations as it compensates the contribution from the ABJ anomaly. Second, the constraint $da \wedge d\tilde{b} = 0$ makes the theory invariant under $U(1)_a$ gauge transformations. This is a new type of gauge theory we call non-invertible gauge theory, since it arises from gauging a non-invertible symmetry.

## 3.2 Non-invertible $\tilde{U}(1)$ gauge theory: $E \cdot B = 0$ QED

Consider a theory with one Weyl fermion. The $U(1)$ transformation acting as $\psi \to e^{i\alpha}\psi$ suffers from a triangle anomaly and a gravitational anomaly and cannot be gauged. Can we gauge a non-invertible version of it? For the purpose of the exercise we will ignore the gravitational anomaly. It is also possible to cancel it by adding extra fermions but it won't be important for this discussion. This is almost equivalent to the previous example, just that we have only one $U(1)$ and one gauge field instead of two. This symmetry can be gauged using the same prescription. The scalar $\phi$ is inserted into the Lagrangian without any kinetic term or potential. It only appears as a Lagrange multiplier enforcing $d\tilde{a} \wedge d\tilde{a} = 0$. The difference now is that the constraint involves only the gauge field $\tilde{a}$. This is because the constraint is needed to cancel the triangle $U(1)^3$ anomaly. The $\tilde{U}(1)$ non-invertible gauge theory can be written as

$$\mathcal{L} = \frac{1}{4g^2}(d\tilde{a})^2 + i\psi^\dagger \sigma^\mu D_\mu \psi \tag{28}$$

together with the $d\tilde{a} \wedge d\tilde{a} = 0$ constraint. Notice that the extra term $\tilde{a} \wedge \tilde{a} \wedge d\tilde{a}$ is identically zero due to antisymmetry of the indices. We see that a non-invertible $\tilde{U}(1)$ gauge theory is equivalent to an ordinary $U(1)$ with the constraint $E \cdot B = 0$.

# 4 't-Hooft anomalies for non-invertible symmetries

To study 't-Hooft anomalies we need to couple the non-invertible symmetry to a background gauge field. The procedure is very similar to the gauging studied above, just that we don't integrate over the value of the gauge field. The same constraints however still apply. As an example consider a theory of 2 Weyl fermions with the $U(1)_a \times U(1)_b$ charges,

$$q_1 = (1, 1), \ q_2 = (-1, 1). \tag{29}$$

$U(1)_a$ is anomaly free and can be gauged dynamically. Before the $U(1)_a$ gauging, $U(1)_b$ was an exact symmetry with certain 't-Hooft anomalies. In particular, there is a triangle $U(1)_b^3$ anomaly, and a $U(1)_b \times gravity$ anomaly, both with coefficients $1^3 + 1^3 = 1 + 1 = 2$. However, there is also a mixed anomaly of the form $U(1)_a^2 \times U(1)_b$. Due to this anomaly, the $U(1)_a$ gauging breaks $U(1)_b$ explicitly. $U(1)_b$ is not a symmetry of the theory and therefore doesn't

give rise to rigorous anomaly matching conditions. However, this observation raises a question. $U(1)_b$ is not a symmetry, but its non-invertible version $\tilde{U}(1)_b$ is an exact symmetry which (at least naively) possesses the same anomalies. But it seems like the anomalies for $\tilde{U}(1)_b$ are not matched, on the same way anomalies for $U(1)_b$ are not matched. Not surprisingly, the resolution comes from the differences between $U(1)_b$ and $\tilde{U}(1)_b$ and in particular the constraint $d\tilde{b} \wedge da = 0$. We will start by showing explicitly the lack of $U(1)_b^3$ "anomaly" matchings by adding and condensing a scalar. Later, we will show how the constraint leads to matching of the $\tilde{U}(1)_b^3$ anomaly.

We can see the lack of $U(1)_b^3$ "anomaly" matching conditions in the following way. Deform the theory by adding a scalar $\chi$ with $U(1)_a \times U(1)_b$ charges of $q_\chi = (2, -2)$, and the Yukawa interaction $\chi \psi_2 \psi_2 + c.c.$. We can give it a vev of the form $\langle \chi \rangle = v$. As a result $U(1)_a$ is Higgsed, and $U(1)_b$ is locked with $U(1)_a$ gauge transformations. The fermion $\psi_2$ gets a mass from the Yukawa term, and the low energy theory consists of only $\psi_1$ with charge $q_b = 2$ due to the color-flavor locking pattern.[3] This low energy theory does match the $U(1)_b \times gravity$ anomaly of the uv theory, but it doesn't match the $U(1)_b^3$ anomaly. This is fine because $U(1)_b$ is not a symmetry of the theory. What happens if instead of considering the anomalous $U(1)_b$, we consider its non-invertible version $\tilde{U}(1)_b$? Now this is an exact symmetry of the theory and is expected to give rise to anomaly matching conditions. In particular, the same $\tilde{U}(1)_b^3$ anomaly still exists. When coupled to a $\tilde{U}(1)_b$ background gauge field, under $\tilde{U}(1)_b$ transformations the action is shifted by

$$\delta S \sim \int d\tilde{b} \wedge d\tilde{b}. \tag{30}$$

How is it matched in the infrared? To study the $\tilde{U}(1)_b^3$ anomaly, we should couple the symmetry to a background gauge field $\tilde{b}$, with non-trivial value for $\int d\tilde{b} \wedge d\tilde{b}$. As explained above, when coupling to a non-invertible gauge field, we must impose the constraint $da \wedge d\tilde{b} = 0$. This is true even when $\tilde{b}$ is a background gauge field, because $\phi$ is dynamical. Consider again deforming the theory by adding the same Higgs field $\chi$ as before. $U(1)_a$ is again Higgsed, and $\psi_2$ gets a mass from the Yukawa term. Naively, we get at low energies only the fermion $\psi_1$ but this fermion is not enough to match the $\tilde{U}(1)_b^3$ anomaly. The resolution is that the vev of $\chi$ in this case cannot be simply a constant $\langle \chi \rangle = v$. To see it we can look at the kinetic term for $\chi$:

$$|\partial_\mu \chi - 2ia_\mu \chi + 2i\tilde{b}_\mu \chi|^2. \tag{31}$$

We can try and plug in $\chi = v$ and take $a = \tilde{b}$ to minimize the kinetic term. However, this is forbidden. The reason is that plugging $a = \tilde{b}$ into the constraint $da \wedge d\tilde{b} = 0$, implies $d\tilde{b} \wedge d\tilde{b} = 0$ which is inconsistent with our choice of background. To minimize the kinetic term, the vacuum configuration for $\chi$ in this background must support a vortex on each one of the 2d manifolds on which $\int d\tilde{b} \neq 0$. Therefore, the low energy theory is not just $\psi_1$, but $\psi_1$ together with two orthogonal vortices. These vortices support fermion zero modes and the whole system together matches the $\tilde{U}(1)_b^3$ anomaly. We would like to make several comments on the non-invertible anomaly matching:

- The philosophy behind anomaly matching is that given a flow from a uv theory $\mathcal{T}_{uv}$ to an IR theory $\mathcal{T}_{IR}$, the two theories must have the same anomalies. Importantly, we can first flow to $\mathcal{T}_{IR}$ and then couple it to background gauge fields and we will get the same result as if we would first couple to gauge fields and then flow to the IR. The reason is that we can always choose the background to be very weak such that it doesn't affect the local dynamics until the end of the flow. For the non-invertible $\tilde{U}(1)_b^3$ anomaly studied here this doesn't work. We must first couple to background gauge fields and

---

[3]There is also a residual $\mathbb{Z}_2$ gauge symmetry but it won't be relevant for our discussion.

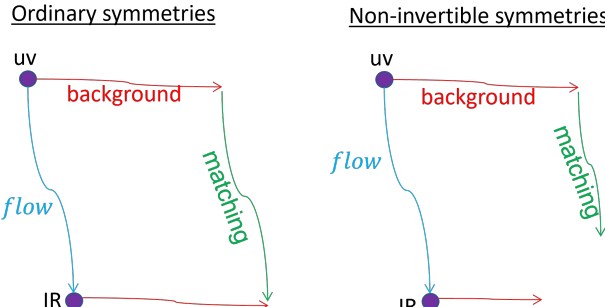

**Figure 2:** Formally, anomalies are matched along the green line. Only after we couple to gauge fields we can probe the anomaly and derive rigorous consistency conditions. If $[flow, background] = 0$ as for ordinary symmetries, we can pull back the matching to the blue line that connects the uv and the IR without coupling to gauge fields. When $[flow, background] \neq 0$ as was the case for the $\tilde{U}(1)_b^3$ anomaly, the matching occurs only along the green line and cannot be pulled back to the blue line.

then flow. The reason is that background gauge fields for the non-invertible symmetry impose constraints and can change the dynamics of the theory drastically, even if the background it self is very weak. We can summarize this by saying that RG flow and background gauging don't commute:

$$[\text{flow}, \text{background}] \neq 0 \,. \tag{32}$$

See figure 2 for more details.

- We saw that the $\tilde{U}(1)_b \times gravity$ anomaly is matched by the effective theory consisting only $\psi_1$. The reason is that the background gauging needed to study this anomaly involves the metric. This background gauging does commute with RG flow and therefore this anomaly must be matched as in the ordinary case. This anomaly matching is not accidental as one might think by looking only at the anomalous axial transformation, but is a consequence of the exact non-invertible symmetry.

- The importance of the constraint $da \wedge d\tilde{b} = 0$ is manifest in this procedure. Without the constraint, we could have simply solve the vacuum equations by setting $a = \tilde{b}$ without having a vortex and violating anomaly matching conditions.

- Getting a vortex at low energies due to some non-trivial background $U(1)$ gauge field is very common when the $U(1)$ symmetry is spontaneously broken. If a $U(1)$ is spontaneously broken, the vacuum equations for the condensate are solved by a vortex configuration exactly as in our case. The main difference is that in our case there is no Goldstone boson. There is a condensate $\chi$ but this condensate doesn't break the $\tilde{U}(1)_b$ global symmetry thanks to the color-flavour locking pattern. Once we turn on a non-trivial background for $\tilde{b}$, the color-flavour locking pattern is not allowed due to the constraint $da \wedge d\tilde{b} = 0$ and we get a vortex configuration as if the symmetry was broken.

- Concretely, the mentioned non-invertible anomaly matching condition doesn't constrain the low energy theory with $\tilde{b} = 0$. It does constrain the effective theory in this background which happens to involve a soliton configuration. Therefore, we can say that the anomaly matching conditions constrain the effective theory on the soliton. These conditions are satisfied thanks to the fermionic zero modes living on the vortex [23].

# 5  2d U(1) and 4d SU(N) examples

So far we focused on $U(1)$ gauge theories in 4d. Can this procedure be generalized to other cases? In particular, we will comment on two important cases where an anomalous $U(1)$ symmetry famously exists, and discuss the obstruction of lifting it to a non-invertible symmetry.

## 5.1  2d U(1) gauge theory

Consider a 2d $U(1)_a$ gauge theory with one Dirac fermion of charge $q = 1$. Similarly to the 4d case, we have the axial $U(1)_b$ with the anomalous current $d \star j_b = \frac{1}{2\pi} da$. Can we promote this symmetry to a non-invertible symmetry? naively all we need to do is redefine the current to be

$$\star \tilde{J}_b = \star j_b - \frac{1}{2\pi} a + \frac{1}{2\pi} d\phi \ . \tag{33}$$

The current is conserved, and the operator

$$\int \mathcal{D}\phi \exp\left( i\alpha \int_{\mathcal{M}_1} \star \tilde{J}_b \right) = \exp\left( i\alpha \int_{\mathcal{M}_1} \left[ \star j_b - \frac{1}{2\pi} a \right] \right) \int \mathcal{D}\phi \exp\left( i\alpha \int_{\mathcal{M}_1} \frac{1}{2\pi} d\phi \right) \tag{34}$$

is gauge invariant and topological. However, the integration over $\phi$ can be replaced by a sum over its holonomies around $\mathcal{M}_1$ which results in

$$\int \mathcal{D}\phi \exp\left( i\alpha \int_{\mathcal{M}_1} \frac{1}{2\pi} d\phi \right) = \sum_k \exp(i\alpha k) \ . \tag{35}$$

This is identically zero unless $\alpha \in 2\pi\mathbb{Z}$. We see that the operator is equivalent to the 0 operator. 0 is of course gauge invariant, topological and non-invertible but it is not good for anything. We see unfortunately that the generalization to 2d gauge theories fails.

## 5.2  4d $SU(N)$ gauge theory

Similar thing happens for the case of 4d $SU(N)$ gauge theories. Consider an $SU(N)$ gauge theory with fermions and an anomalous $U(1)_b$ current satisfying

$$d \star j_b = \frac{1}{8\pi^2} tr(f \wedge f). \tag{36}$$

Here $f = da - ia \wedge a$ is the $SU(N)$ field strength. When we had a $U(1)$ gauge theory, we had to introduce a $U(1)$ sigma model (i.e. the compact scalar $\phi$) to compensate over the lack of gauge invariance. When dealing with an $SU(N)$ gauge theory, we need to introduce an $SU(N)$ non-linear sigma model parametrized by $W \in SU(N)$. We can define the covariant derivative $DWW^\dagger = dWW^\dagger - ia$. Then, we can write a gauge invariant conserved current of the form

$$\begin{aligned}
\star \tilde{J}_b &= \star j_b - \frac{1}{24\pi^2}[(DWW^\dagger)^3 + 3iDWW^\dagger \wedge f] \\
&= \star j_b - \frac{1}{8\pi^2}(a \wedge da - 2i/3a^3) - \frac{1}{24\pi^2}(dWW^\dagger)^3 - \frac{i}{8\pi^2}d(a \wedge dWW^\dagger).
\end{aligned} \tag{37}$$

It is gauge invariant by construction, and conserved since

$$d \star \tilde{J}_b = d \star j_b - \frac{1}{8\pi^2} tr(f^2) = 0. \tag{38}$$

Using this current, we can define the operator $\tilde{U}_\alpha(\mathcal{M}_3) = \mathcal{D}We^{i\alpha \int \star \tilde{J}_b}$ as before which is topological and gauge invariant. However, we get the same problem as in the 2d case. As part of

the definition of the operator, we must integrate over all values of $W \in SU(N)$. In particular, the contribution from $\frac{1}{24\pi^2}\int (dW W^\dagger)^3$ can be replaced by a sum over integers. Again we find that this operator is proportional to $\sum_k e^{i\alpha k}$ which vanishes unless $\alpha \in 2\pi\mathbb{Z}$.[4] Also this operator seems to simply act as 0 and the procedure again fails.

# 6 Non-invertible symmetries Vs anomalies for anomalous symmetries

In this last section we want to make several comments on some of the similarities between two seemingly different approaches to the anomalous $U(1)$ symmetry and promoting it into an exact symmetry. One approach is the one introduced here, and was also studied in [13, 14]. In this approach we promote the anomalous symmetry to a non-invertible symmetry by adding new degrees of freedom on the topological operator. The second approach is the one taken in [20]. In that approach we promote a discrete $\mathbb{Z}_N$ subgroup of the anomalous symmetry to an exact ordinary symmetry by adding new degrees of freedom to the theory. The dynamics triggered by the new degrees of freedom is such that the symmetry is spontaneously broken. In every vacuum, the effective theory is the original theory of interest. The ABJ anomaly in this setup is a consequence of the spontaneous breaking of the symmetry by the extra degrees of freedom. In this setup, instead of the topological operator $\tilde{U}$, we have a domain wall connecting two $\mathbb{Z}_N$ vacua. The domain wall lives on some 3-manifold $\mathcal{M}_3$. Since it takes us from one $\mathbb{Z}_N$ vacuum to the other, it acts exactly as the anomalous symmetry. The anomaly is compensated by the new degrees of freedom that come back to life on the domain wall. These degrees of freedom contain an emergent Chern-Simons gauge field, similar to the one that was introduced in [13, 14]. In the two approaches, we have a 3d operator acting as axial rotations with new degrees of freedom living on it. The domain wall is not topological due to its tension (it costs energy to deform it), but it is possible to formally define an operator which is the domain wall divided by its tension×volume. This operator is topological and gives an alternative promotion of the anomalous symmetry to an exact one. Effectively, in the two approaches we add new degrees of freedom that live only on the 3d operator. These degrees of freedom are there to cancel the ABJ anomaly. No matter what exactly the details of the new degrees of freedom are, the 3d operator acts the same on the physical bulk degrees of freedom.

There are more similarities between the two approaches that we want to point out. In [20], it was shown that by adding the degrees of freedom as described above, one can get an effective Yang-Mills theory with $2\pi/N$ periodicity for the theta angle. As a result, there is a time reversal symmetry at $\theta = \pi/N$. Microscopically, this is because the heavy fields jump from one vacuum to the other as we cross $\theta = \pi/N$. At $\theta = \pi/N$ there is a 2-fold vacuum degeneracy due to the spontaneous breaking of time reversal. See section (2) of [20] for more details. Similarly, in [24] it was argued that for pure Maxwell theory, there is a non-invertible time reversal symmetry for $\theta = \pi/N$. Another similarity between the two approaches is that the anomalies for the anomalous symmetry cannot be used to constrain the vacuum of the theory but can be used to constrain the effective theory on dynamical solitons. This is explained in section 4 here and in section (4) of [20] for the two approaches respectively. There are also some differences. For example, the non-invertible approach seems to work only for abelian gauge theories, while the second approach works very well also for non-abelian ones. On the

---

[4]This case is a more subtle because $W$ appears also inside the term $d(a \wedge dW W^\dagger)$. This term is a total derivative and on simple enough spaces, doesn't contribute. Hence, we get $\tilde{U}_\alpha = 0$. There might be scenarios where the term $d(a \wedge dW W^\dagger)$ saves us from getting $\tilde{U}_\alpha = 0$. It will be very interesting to explore this and see if the restrictions are not too strong and one can use $\tilde{U}_\alpha$ to learn something interesting.

other hand, using the non-invertible approach we can save the entire $U(1)$, while using the second approach only a discrete $\mathbb{Z}_N$ subgroup can be saved. Even though the two methods seem to have a very different origin, it looks like the physical consequences of having them is similar. Understanding better the relation between promoting the anomalous symmetry to a non-invertible symmetry and promoting it to a spontaneously broken symmetry, can shed new light on non-invertible symmetries and help find new ones. We hope to pursue in this the direction in the future.

# Acknowledgements

We would like to thank Masazumi Honda, Rishi Mouland, Kaan Onder, Shu-Heng Shao, Tin Sulejmanpasic, and David Tong for fruitful discussions.

**Funding information** We are supported by the STFC consolidated grant ST/P000681/1 and the EPSRC grant EP/V047655/1 "Chiral Gauge Theories: From Strong Coupling to the Standard Model".

# A  The factor of 2

In this appendix we will explain in detail the factor of 2 that appears in equation (15). Consider the following action,

$$S = \frac{\alpha}{8\pi^2} \int_{S^1 \times S^2} (a - d\phi) \wedge da \,, \tag{A.1}$$

where $a$ is a $U(1)$ gauge field and $\phi$ is a compact scalar subject to the gauge transformations,

$$a \to a + d\omega \,, \quad \phi \to \phi + \omega \,. \tag{A.2}$$

If we plug in $\phi$ such that $\int_{S^1} d\phi = 2\pi n$, and $a$ such that $\int_{S^2} da = 2\pi m$ naively we get

$$S \to \frac{\alpha}{8\pi^2} \int_{S^1 \times S^2} a - \frac{\alpha k m}{2} \,. \tag{A.3}$$

Our claim that leads to the result in (15) is that the correct result is actually

$$S \to \frac{\alpha}{8\pi^2} \int_{S^1 \times S^2} a - \alpha k m \,. \tag{A.4}$$

One consistency condition is that when $\alpha \in 2\pi\mathbb{Z}$, the CS part of the action is gauge invariant by itself and it is expected that the scalar term will be trivial. This is true only when the factor of two is included. Below, we will give a more direct derivation of this mysterious factor of two. To understand this, we will first review a related factor of two that appears in the pure Chern-Simons action,

$$S_{CS} = \frac{k}{4\pi} \int a \wedge da \,. \tag{A.5}$$

Naively, the gauge variation of the CS action is

$$\delta S_{CS} = \frac{k}{4\pi} \int d\omega \wedge da \in \pi k \mathbb{Z} \,, \tag{A.6}$$

since $\int d\omega \in 2\pi\mathbb{Z}$ and $\int da \in 2\pi\mathbb{Z}$. This implies that the action is gauge invariant only if $k$ is even. This naive derivation is wrong. The first thing we need to do is to define the action in a non-ambiguous way. On topologically non-trivial manifolds, $a$ cannot be globally well defined and a way to continue is to define the integral on patches. To simplify things, we will consider a concrete example where we take the manifold to be $S^1 \times S^2$. For a configuration with non-zero monopole flux, $a$ cannot be defined smoothly on the entire sphere, but we can divide the sphere to two patches denoted by $H_N$ and $H_S$ with $a_{N,S}$ defined globally on each patch. We will denote the intersection between the two patches by $E$. On the intersection, $a_N - a_S = d\lambda$ where $\lambda$ is some periodic scalar. The first attempt to define the action is to write it as

$$S_{naive} = \frac{k}{4\pi} \int_{S^1} \int_{H_N} a_N \wedge da + \frac{k}{4\pi} \int_{S^1} \int_{H_S} a_S \wedge da. \tag{A.7}$$

However, this action depends on the arbitrary choice of patches. The way to correct it is to add a boundary term on the intersection such that

$$S = \frac{k}{4\pi} \int_{S^1} \int_{H_N} a_N \wedge da + \frac{k}{4\pi} \int_{S^1} \int_{H_S} a_S \wedge da + \frac{k}{4\pi} \int_{S^1} \int_{E} d\lambda \wedge a. \tag{A.8}$$

To see the problem with (A.7) explicitly, denote the coordinates on the $S^1$ by $\tau \in [0, 2\pi]$ and the coordinates on the sphere by the usual $[\theta, \varphi]$, and take the following configuration

$$a_\tau^N = p, \ a_\tau^S = 0, \ a_\theta = 0, \ a_\varphi^N = \frac{m(1 - cos(\theta))}{2sin(\theta)}, \ a_\varphi^S = \frac{m(-1 - cos(\theta))}{2sin(\theta)}, \ \lambda = p\tau + m\varphi. \tag{A.9}$$

We can choose the intersection $E$ to lie on some $\theta = \theta_0$ circle. An explicit computation shows that (A.7) is equal to

$$S_{naive} = \frac{kpm}{8\pi} \int_0^{2\pi} d\tau \int_0^{\theta_0} d\theta sin(\theta) \int_0^{2\pi} d\varphi = \frac{\pi kpm}{2}(1 - cos(\theta_0)). \tag{A.10}$$

The result depends on the arbitrary choice of $\theta_0$ which doesn't make any sense. Therefore, (A.7) is not a good definition of the action. On the other hand, the contribution from the boundary integral is

$$\frac{k}{4\pi} \int_{S^1} \int_{E} d\lambda \wedge a \equiv \frac{k}{4\pi} \int_0^{2\pi} d\tau \int_0^{2\pi} d\varphi sin(\theta_0) \left( \frac{1}{sin(\theta_0)} \partial_\phi \lambda a_\tau - \partial_\tau \lambda a_\varphi \right)$$
$$= \frac{\pi kpm}{2}(1 + cos(\theta_0)). \tag{A.11}$$

Together, we get $S = \pi kpm$ which is independent of $\theta_0$ as required. As another example, consider the configuration,

$$a_\tau^N = a_\tau^S = c, \quad a_\theta = 0, \quad a_\varphi^N = \frac{m(1 - cos(\theta))}{2sin(\theta)}, \quad a_\varphi^S = \frac{m(-1 - cos(\theta))}{2sin(\theta)}, \quad \lambda = m\varphi. \tag{A.12}$$

The action is

$$S = \pi kcm + \pi kcm = 2\pi kcm. \tag{A.13}$$

Importantly, we get two equal contributions. One from the bulk integral and one from the boundary integral. The full action is invariant mod $2\pi$ under the gauge transformation $c \to c + 1$, given that $k \in \mathbb{Z}$. As we see, the boundary term is crucial to get the correct normalization of the level.

Our next step is to write explicitly the 3d CS action coupled to a compact scalar as in (12). We add the compact scalar $\phi$ to the theory that transforms under gauge transformations $a \to a + d\omega$ as $\phi \to \phi + \omega$. We claim that the correct way to write $\int (a - d\phi) \wedge da$ is[5] (on $S^1 \times S^2$ as a concrete example)

$$
\frac{\alpha}{8\pi^2} \int_{S^1 \times S^2} (a - d\phi) \wedge da \equiv \frac{\alpha}{8\pi^2} \int_{S^1} \int_{H_N} (a_N - d\phi_N) \wedge da + \frac{\alpha}{8\pi^2} \int_{S^1} \int_{H_S} (a_S - d\phi_S) \wedge da
$$
$$
+ \frac{\alpha}{8\pi^2} \int_{S^1} \int_E d\lambda \wedge (a - d\phi).
$$
(A.14)

As in (15) we will take a configuration where $\phi$ winds around the $S^1$, $\int_{S^1} d\phi = 2\pi k$, and get

$$
\frac{\alpha}{8\pi^2} \int_{S^1 \times S^2} d\phi \wedge da \equiv \frac{\alpha}{8\pi^2} \int_{S^1} \int_{S^2} d\phi \wedge da + \frac{\alpha}{8\pi^2} \int_{S^1} \int_E d\lambda \wedge d\phi
$$
$$
= \frac{\alpha k}{4\pi} \int_{S^2} da + \frac{\alpha k}{4\pi} \int_E d\lambda = \alpha k m,
$$
(A.15)

where $m$ is again the magnetic flux. We see that also here we get two equal contributions, one from the bulk integral and one from the boundary integral, resulting in the factor of 2 mentioned in (15).

## A.1 The BF theory

While it is not directly related to the main body of the text, it might be useful to write the analogue of (A.8 ) for the coupling between two gauge fields, $a, b$, also known as the BF theory. The action is

$$
S = \frac{k}{2\pi} \int_{S^1 \times S^2} a \wedge db \equiv \frac{k}{2\pi} \int_{S^1} \int_{H_N} a_N \wedge db + \frac{k}{2\pi} \int_{S^1} \int_{H_S} a_S \wedge db + \frac{k}{2\pi} \int_{S^1} \int_E d\lambda_a \wedge b_N.
$$
(A.16)

Notice that in the last term we wrote $b_N$ for concreteness but equivalently we can use $b_S = b_N - d\lambda_b$ instead. The difference between the two choices is trivial,

$$
\Delta S = \frac{k}{2\pi} \int_{S^1} \int_E d\lambda_a \wedge d\lambda_b \in 2\pi\mathbb{Z}.
$$
(A.17)

---

[5]In fact, this is not the end of the story. The integrals over $S^1$ and $E$ requires some refinement in the spirit of equation (2.6) of [25]. This will lead to extra 2d, 1d and 0d integrals on the intersections of the various patches. However, for simplicity we ignore these extra terms as they are not needed for the specific result derived here.

This form of the action has several nice properties. First, it is symmetric under the exchange of $a \leftrightarrow b$, since

$$
\begin{aligned}
\frac{k}{2\pi} \int_{S^1 \times S^2} a \wedge db &\equiv \frac{k}{2\pi} \int_{S^1} \int_{H_N} a_N \wedge db + \frac{k}{2\pi} \int_{S^1} \int_{H_S} a_S \wedge db + \frac{k}{2\pi} \int_{S^1} \int_E d\lambda_a \wedge b_N \\
&= \frac{k}{2\pi} \int_{S^1} \int_{H_N} [da_N \wedge b_N - d(a_N \wedge b_N)] \\
&\quad + \frac{k}{2\pi} \int_{S^1} \int_{H_S} [da_S \wedge b_S - d(a_S \wedge b_S)] + \frac{k}{2\pi} \int_{S^1} \int_E d\lambda_a \wedge b_N \\
&= \frac{k}{2\pi} \int_{S^1} \int_{H_N} b_N \wedge da + \frac{k}{2\pi} \int_{S^1} \int_{H_S} b_S \wedge da + \frac{k}{2\pi} \int_{S^1} \int_E [d\lambda_a \wedge b_N - a_N \wedge b_N + a_S \wedge b_S] \\
&= \frac{k}{2\pi} \int_{S^1} \int_{H_N} b_N \wedge da + \frac{k}{2\pi} \int_{S^1} \int_{H_S} b_S \wedge da + \frac{k}{2\pi} \int_{S^1} \int_E d\lambda_b \wedge a_S \equiv \frac{k}{2\pi} \int b \wedge da \, .
\end{aligned}
$$
(A.18)

A second property is the manifestation of the $(\mathbb{Z}_k)_a \times (\mathbb{Z}_k)_b$ global 1-form symmetries, acting as $a \to a + \frac{1}{k} d\omega_a$ , $b \to b + \frac{1}{k} d\omega_b$.

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
