# Peer review of "On anomalies and gauging of U(1) non-invertible symmetries in 4d QED"

_SciPost Physics, doi:SciPost Phys. 15, 002 (2023)_

## Round 2 · Referee Report · Anonymous (Referee 1) · 2023-1-25

Report

The manuscript concerns with a classical $U(1)$ symmetry which suffers from an abelian ABJ anomaly, and discusses a way to promote it to an exact non-invertible global symmetry. This had been achieved in the previous literature only for rational rotation angles, and the author proposes an alternative construction where both rational and irrational rotation angles can be made into a non-invertible symmetry. The author then discusses how to couple the non-invertible symmetry to either a dynamical or background gauge field. In the latter case, the corresponding ‘t Hooft anomaly matching condition is carefully discussed with an example. The author also discusses similar approaches for two-dimensional abelian ABJ anomalies and four-dimensional non-abelian ABJ anomalies and explains why in those cases one does not obtain new non-invertible global symmetries. Finally, a potential relation to the author’s previous work on “anomalies for anomalous symmetries” is mentioned.

The proposed non-invertible topological operator is constructed by summing over the gauge orbit of a naïve topological operator which is not gauge-invariant. The gauge parameter is promoted to a dynamical compact scalar field $\phi$ living on the non-invertible operator.

The referee is mainly concerned by the following:
- The explanation as to why the proposed operator is topological feels a bit too quick. The main argument is that the current in (2.4) is conserved. On the one hand, this is not a necessary condition for the operator to be topological. For instance, the non-invertible topological operator from the previous literature (shown in (2.15) of the manuscript) does not satisfy this property. On the other hand, it is also not immediately obvious that this is a sufficient condition. Since the current (2.4) depends on the scalar field $\phi$ which lives only on the non-invertible operator, it is hard to see that how the conservation equation can be used to move around the operator using the Stokes’ theorem as usual. At the minimum, the referee feels that the scalar field $\phi$ should first be extended to the higher dimensional bulk for an argument along this line to work.

However, even though there is such a subtlety in the argument, the author’s following demonstration that, for rational angles, the proposed operator reduces to the same non-invertible topological operator previously known in the literature, and the fact that it is possible to consistently couple to gauge fields for the proposed non-invertible symmetry, seem to suggest that the proposed operator is indeed topological. Also, it is indeed an important question to address how to gauge non-invertible symmetries and to understand their ‘t Hooft anomalies, and the current manuscript makes a step toward this urgent research direction. Moreover, the author makes an interesting observation that even turning on a non-dynamical background gauge field for a non-invertible symmetry may affect the dynamics of the system significantly.

Therefore, the referee believes that the manuscript merits publication in SciPost, if the above concern can be addressed at least briefly. Below is one minor point:
- The description of an ordinary ‘t Hooft anomaly around (1.1) is imprecise. The functional $S[A]$ should be a functional of both the background gauge field and the gauge parameter. The paragraph below (1.1) should be edited accordingly, for instance the sentence below (1.1) is self-contradictory.
  • validity: -
  • significance: -
  • originality: -
  • clarity: -
  • formatting: -
  • grammar: -

Author:  Avner Karasik  on 2023-03-06  [id 3438]

(in reply to Report 1 on 2023-01-25)

Dear referee,
Thank you for the report.
I will also elaborate about it in the manuscript, but just to explain about the topological nature of the defect.
First, if phi was a 4d field, then one can use Stokes theorem and in this case, the fact that the current is conserved is a sufficient condition for proving that the operator is topological.
phi is defined only on the 3d manifold but if we are able to extend it to the bulk, we can use an extension and prove that the operator is topological.
Can we do it? In general, it will not be possible to extend phi to the entire 4d spacetime without singularities (because maybe phi winds around a contractible S1 for example).
However, this is not needed.
When we continuously deform the manifold M3 to some M3' we cover a 4d manifold with the topology of M3X I, where I is an interval. All we need is to be able to extend phi from M3 to M3X I and this is always possible, because the topology of M3 is preserved along the deformation.
Therefore, we can extend it at least to this 4d submanifold, use Stokes theorem and prove that the operator is topological.

Best,
Avner

---

## Round 2 · Referee Report · Anonymous (Referee 2) · 2023-2-1

Report

A very active topic of research in the last couple of years concers the extension of symmetry operators to categorical symmetries, including non-invertible symmetries. In this paper the author introduced a new construction of non-invertible symmetries for theories with U(1) currents that are not conserved in a mild way. The requisite is that the non-conserved current can be improved to a conserved but not gauge invariant current. The author proposes a topological operator associated with this current such that it can be defined for any U(1) angle, in contrast with previous constructions in the literature, which are only defined for rational values.

The main insight is that, given a gauge-variant current, one may sum over a gauge orbit to obtain a gauge invariant current. In concrete examples this is achieved by introducing Stueckelberg-like fields that compensate the gauge transformation of the gauge field.

The paper also contains applications of this idea such as characterizing the anomalies of these non-invertible symmetries and their gauging, which is an important open problem.

It is an interesting work that makes progress towards the understanding of symmetries in these theories. I recommend publication if the following issues are adressed.

  1. The author makes the would-be topological defects gauge invariant by introducing a compact Stueckelberg-like scalar field $\phi$ that transforms under gauge transformations of the bulk gauge field. It is not clear how such gauge transformation, which is naturally defined over all spacetime, does not spoil the 3d nature of $\phi$. Said in other words, it is not clear that $\phi$ can be restricted to lie in the defect. I think the author should better adress this point. This is related to the caption of Figure 1, where one must indeed extend $\phi$ to the bulk when coupling to background fields.

  2. As the previous referee notes, the justification of the topological nature of the defects could be improved.

  3. In footnote 1 the author mentions a tricky factor of 2. The author should further elaborate on this point, as it is not clear to me whether that factor is correct.

Requested changes

  1. Explain in greater detail how $\phi$ is restricted to the topological operator.

2.Further justification of the topological nature of the operator.

  1. Correct the factor of 2 mentioned in footnote 1.

  • validity: -
  • significance: -
  • originality: -
  • clarity: -
  • formatting: -
  • grammar: -

Author:  Avner Karasik  on 2023-03-06  [id 3441]

(in reply to Report 2 on 2023-02-01)

Dear referee,

Thank you for the detailed report. I would like to address the 3 points you mentioned:

  1. I am not sure I understand the subtlety. ϕ transforms under gauge transformations locally as ϕ ->ϕ+w. Why would the gauge transformation spoil the 3d nature of ϕ? Probably I misunderstood the issue you are raising and I will be happy if you can elaborate a little bit more.
  2. This is a good point. I will also elaborate about it in the manuscript, but just to explain about the topological nature of the defect. First, if ϕ was a 4d field, then one can use Stokes theorem and in this case, the fact that the current is conserved is a sufficient condition for proving that the operator is topological. phi is defined only on the 3d manifold but if we are able to extend it to the bulk, we can use an extension and prove that the operator is topological. Can we do it? In general, it will not be possible to extend phi to the entire 4d spacetime without singularities (because maybe ϕ winds around a contractible S1 for example). However, this is not needed. When we continuously deform the manifold M3 to some M3' we cover a 4d manifold with the topology of M3X I, where I is an interval. All we need is to be able to extend phi from M3 to M3X I and this is always possible, because the topology of M3 is preserved along the deformation. Therefore, we can extend it at least to this 4d submanifold, use Stokes theorem and prove that the operator is topological.
  3. I wrote a very detailed appendix that explains the factor of 2. It will appear in the new version that I hope to send in a couple of days.

Thank you very much. Best, Avner

---

## Round 2 · Referee Report · Nabil Iqbal (Referee 3) · 2023-2-26

Strengths

See below.

Weaknesses

See below.

Report

Overall, this is an extremely interesting paper, and one that I feel is very important. There is much to think about here; it presents a different way to think about chiral non-invertible symmetry; the defect constructed here is really quite different from those in earlier work, and in many ways it is simpler and leads to generalizations that are not obvious in the earlier constructions. This includes the possibility of gauging the non-invertible symmetry (at the cost of having a somewhat different bulk dynamics) and finding interesting new generalizations of the idea of anomaly cancellation. I believe that it should be published. 

There are however some points which I found confusing, and I think some further explanation would improve the presentation of the paper.

I have ordered my comments below in order of importance. I think really only the first two are essential.

  1. It is argued that the defect here is essentially the same as that of earlier work by Choi et. al and Cordona et. al if the rotation angle \alpha is rational. This (in my opinion) a rather strong statement and I feel that it needs to be justified very carefully. In the process of doing this, below (2.16), it is argued that if the flux of da is nonzero modulo (2\pi N) then there is no solution to the classical equation of motion for dA = -\frac{1}{N} da and thus the partition function of D_{\frac{2\pi}{N}} must vanish. It is not entirely clear to me that the non-existence of a classical saddle for A means that the entire quantum partition function must vanish; given its importance for the remainder of the argument, could this be justified somewhat more carefully? (For what it’s worth, I do believe the statement is correct). I would particularly appreciate seeing a more detailed justification for the case of nontrivial rational \alpha where it is simply stated that the argument holds the same way (and where the TQFT involved is somewhat less familiar).

  2. It is stated in the same paragraph that it is not clear that the operator (2.6) can be normalized correctly to to give rise to a well-defined topological defect (emphasis on the word defect, with credit given to Shu-Heng Shao). Could it be explained more clearly what this means? Above (2.8) it is argued (convincingly, I feel) that the operator is topological — and as its a codimension-1 object, it seems it is certainly a “defect” in my understanding of the word — so I couldn’t quite understand what the statement means.

  3. The discussion on gauging the non-invertible symmetry in the bulk is extremely interesting, particularly the effects of the vortices on p13. However one question that I might have is that once the scalar field \phi is allowed to live off of the defect, it seems that one might worry that bulk mass terms for the dynamical photon field might be generated, of the form (a - d\phi)^2, generically Higgsing the bulk photon. It would be nice to see some discussion of whether or not this is expected to happen (and what it means if it does).

  4. A minor comment on wording: on p7, it is argued that only a “discrete” subgroup of the axial U(1) can be restored using the approach of the earlier refs [13, 14]; I’m not sure the word “discrete” is completely appropriate here, as really it is basically all rational values of the rotation angle, and these are dense in U(1) (which is not really what one associates with the word discrete).

  5. A further minor comment: Eq (2.8) is slightly confusing: as \phi lives on the defect, how should one think about the derivatives of \phi (2.8) in directions that are off of the defect? I imagine that (2.8) is meant to hold for any extension of \phi off of the defect? (I think this is definitely true in any case!)

Requested changes

See above.

  • validity: top
  • significance: top
  • originality: top
  • clarity: high
  • formatting: perfect
  • grammar: perfect

Author:  Avner Karasik  on 2023-03-07  [id 3445]

(in reply to Report 3 by Nabil Iqbal on 2023-02-26)

Dear Nabil,

Thank you very much for the detailed report. I will address here the issues you raised above:

  1. The statement that the partition function of the defect vanishes if the magnetic flux is not a product of N, already appears in the paper by Cordova and Ohmori, see equations (29)-(30). A way to show it for general rational angle p/N is from anomalies. The theory on the defect has a ZN 1-form symmetry with anomaly p. The photon field strength da/N plays the role of the ZN 2-form gauge field. A consequence of the anomaly is D_{p/N}=exp(ip/(2\pi N )\int dw^da)D_{p/N}. D_{p/N} doesn't vanish only if the phase is trivial for every w, which is true only if the magnetic flux is a product of N. I will add this argument to the next version of the paper.
  2. The comment in the paper is indeed vague and confusing, and doesn't contribute much to the paper. I think it will best to erase it from the next version. Though, we can still discuss its meaning.
  3. As shown in the paper, phi appears only linearly and can be integrated out. It imposes a constraint and that's it. If you choose not to integrate it out, you might get a dual description in which the scalar gets a kinetic term (and maybe it is worth studying it at some point). However, the simplest thing to do is just to integrate it out at the beginning and forgetting about it.
  4. Agree.
  5. I will also elaborate about it in the manuscript, but the answer is yes. When we continuously deform the manifold M3 to some M3' we cover a 4d manifold with the topology of M3X I, where I is an interval. All we need is to be able to extend phi from M3 to M3X I and this is always possible, because the topology of M3 is preserved along the deformation. Therefore, we can extend it at least to this 4d submanifold, use Stokes theorem and prove that the operator is topological.

Best, Avner

---

## Round 3 · List of Changes

1) Added an appendix to explain a subtle factor of 2 computation. 2) page 7: Explained in detail why the operator in (2.6) is indeed topological. 3) Page 9: Showed that for every rational angle, the previous constructions coincide with the one presented in the paper. 4) Corrected the description of anomalies after (1.1). 5) page 7: deleted the word "discrete" when talking about the group of rational numbers. 6) Deleted the comment about the defects VS operators and their normalization

---

## Editorial Decision

published